# Radon in Schools: A Review of Radon Testing Efforts in Canadian Schools

**DOI:** 10.3390/ijerph18105469

**Published:** 2021-05-20

**Authors:** Sandy Shergill, Lindsay Forsman-Phillips, Anne-Marie Nicol

**Affiliations:** 1CAREX Canada, Simon Fraser University, Burnaby, BC V5A1S6, Canada; sshergill@live.com; 2Faculty of Health Sciences, Simon Fraser University, Burnaby, BC V5A1S6, Canada; 3Faculty of Medicine, McGill University, Montreal, QC H3A0G4, Canada; 4National Collaborating Centre for Environmental Health (NCCEH), Vancouver, BC V5Z4R4, Canada

**Keywords:** radon, schools, Canada, air pollutants, environment

## Abstract

Radon, a known carcinogen, becomes a health risk when it accumulates inside buildings. Exposure is of particular concern for children, as their longer life expectancy increases their lifetime risk of developing cancer. In 2016, 5.5 million students were enrolled in Canadian elementary and secondary schools. With no national policy on radon testing in schools, children may be at risk from radon exposure while attending school and school-based programs. This study explored radon testing efforts in publicly funded Canadian schools and summarizes where testing programs have occurred. Radon testing in schools was identified through a systematic qualitative enquiry, surveying members from different levels of government (health and education) and other stakeholders (school boards, research experts, among others). Overall, this research found that approaches to radon testing varied considerably by province and region. Responsibility for radon testing in schools was often deferred between government, school boards, building managers and construction parties. Transparency around radon testing, including which schools had been tested and whether radon levels had been mitigated, also emerged as an issue. Radon testing of schools across Canada, including mitigation and clear communication strategies, needs to improve to ensure a healthy indoor environment for staff and students.

## 1. Introduction

Radon, referring to radionuclide (^222^Rn), is a tasteless, colourless, odourless gas produced by the decay of uranium naturally present in rock and soil. While outdoor radon levels are generally negligible, radon can permeate building foundations and accumulate to high levels indoors [1]. As a result, over half of overall radiation exposure for individuals comes from radon [2], making it a high public health priority in Canada and internationally [3]. The International Agency for Research on Cancer classifies radon as a carcinogen [4]. Results of research on miners in Canada and around the world clearly establish radon as a human carcinogen [5]. Radon gas emits alpha radiation, and when inhaled, the radiation can directly damage the cells within the lungs [6]. Over time, exposure to high levels of radon gas can increase the risk of developing lung cancer.

In Canada, lung cancer remains the leading cause of cancer death in both men and women, and one of the most commonly diagnosed cancers in Canada. About 86% of lung cancer cases are attributable to modifiable risk factors, such as tobacco use, exposure to radon gas, asbestos, air pollution and certain workplace exposures [7]. Health Canada estimates that over 3000 people die from radon-related lung cancer every year [8]. Exposure to radon gas is the leading cause of lung cancer among non-smokers [8]. Tobacco use and exposure to radon significantly increase the risk of developing lung cancer [8]. As a large portion of lung cancer cases in Canada are attributable to modifiable risk factors, it is also one of the most preventable cancers [7]. Radon testing is particularly important for reducing the risk of radon-induced lung cancer, and for understanding if high levels of radon are present in buildings. However, a 2019 Statistics Canada survey of Canadians reported that only approximately 6% of Canadians had tested for radon in their homes [9].

No national legislation requires radon testing in Canadian public schools, but the federal government recommends testing for radon in homes and all public spaces, which includes schools and childcare centres [10]. While there is no regulation that governs an acceptable level of radon in Canadian homes and buildings, the Government of Canada developed a guideline, in partnership with the Federal Provincial Territorial Radiation Protection Committee, that was adopted in 2007 [10]. The Government of Canada’s radon guideline sets a threshold, based on average annual radon concentration, for when to take action to mitigate radon levels indoors, and this is the same for school buildings at 200 becquerels per cubic metre (Bq/m^3^) [10]. Buildings with levels above this should be remediated to reduce radon exposure. This requires action by a certified professional under the Canadian National Radon Proficiency Program [11] and resources (including funding) to complete remediation efforts.

Exposure to radon is of particular concern for children, as their longer life expectancy increases their lifetime risk of developing cancer [12]. The average ten-year-old spends 21–22 h inside per day [13]. While exposure at home is their most significant source of radon exposure, exposure at school can also contribute to the problem. In the United States, children spend approximately 6.64 h inside school buildings for 180 days per year [14]. In Canada, during the 2016 academic year, approximately 5.5 million students were enrolled in elementary or secondary school programs. As school attendance for youth is often required by national law, exposure to radon within schools is an involutory risk for staff and students [15]. With no national policy to require radon testing in schools, a large number of Canadian children remain at risk of radon exposure [10].

Several American states require radon testing in schools. Three states, Illinois, Iowa, and Tennessee, recommend or suggest radon testing in school buildings [16]. Colorado, Connecticut, Florida, New Jersey, New York, Oregon, Rhode Island, Virginia and West Virginia enforce mandatory radon testing in public schools [16]. Each state’s legislation is regulated by the state government. Legislation differs from one state to another, including private and/or public schools. Some legislation outlines follow-up radon testing protocols after specific intervals, such as every 5 years, or after significant renovations.

Radon testing in schools has also been conducted in Europe. The European Union Council Directive 2013/59/Euratom established safety standards for protection against exposure to ionizing radiation [17]. This led to the creation of the European Research into Radon in Construction Concerted Action (ERRICCA 2) to create policy and legislation to minimize radon exposure. This directive included articles 54 and 103 addressing radon as an occupational exposure, including in school environments. European countries have tested for radon in public school buildings independently in consideration of this directive. Finland, Norway and Switzerland are examples of countries that enforce legal limits for radon levels within school buildings, the reference levels for when to take action to reduce radon in schools varies from 200 to 800 Bq/m^3^ [18]. A comprehensive program for radon testing in schools was also conducted in Ireland from 1998 to 2002. The program tested for radon in 96% of schools in the country [19], where a recommended reference level of 200 Bq/m^3^ and a regulatory reference level of 400 Bq/m^3^ were applied to radon exposure in schools [19].

Testing is the only way to know if radon levels are elevated, and if remediation is needed. Many organizations, including Health Canada or the Government of Canada, provincial health authorities and private companies, are working to raise awareness and address radon exposure in residential homes. The lack of awareness about radon exposure, testing and remediation, coupled with the complexity of who is responsible for ensuring a healthy indoor school environment (is it a public health issue or is it a building issue?) has contributed to less attention paid to school environments. The objectives of this research were to understand the landscape of radon policies and testing practices in public schools in each region of Canada, highlight the variation in school radon policies and testing practices and, based on the findings, identify common themes and opportunities to inform radon policies and practices for Canadian schools going forward.

## 2. Materials and Methods

The research group conducted a systematic qualitative enquiry, surveying members from different levels of government and education, focusing on five key questions (see Table 1).

Together with an introduction letter, the questions were framed around whether or not radon exposure in school environments was being addressed in provinces or territories in Canada, and if related policies, testing data and information could be shared with the research group. As illustrated in Figure 1, using a top-down approach, data collection included participants in one of three groups of contacts as knowledge gatekeepers. The first phase of contacts included ministers and deputy ministers of Health and Education as well as chief medical health officers from each province and territory in Canada. Phase one of data collection began in the spring of 2017.

The second phase of contacts included school board representatives, such as superintendents and chairs of boards within individual school districts or boards. The third phase of contacts included additional knowledge experts and specialists in radon and health care, who may have access to information on regional radon testing in schools. Figure 1 also highlights the response rates from contacts in the first and second phases of data collection.

Contact information for all applicable individuals in each province and territory in Canada was compiled. An introduction letter and the survey were sent by email to phase one participants. We allowed a grace period of three weeks for the first phase contacts to respond. Where there was no response, we contacted the second phase of contacts (school board representatives). Table 1 includes the five open-ended questions that were distributed by email in each phase of data collection. Second phase contacts were given one month to respond. All email correspondences were logged, and detailed answers to each survey question were recorded. Third phase contacts were contacted where details of testing remained unconfirmed or additional information could be provided on testing initiatives. All data collection ended in fall 2017.

Responses to survey questions were categorized. School districts that responded and wished to not participate or did not respond during data collection were categorized as ’testing status unknown’. Regions that indicated testing for radon were recorded as ’tested for radon’, and the exact number of schools included in the testing initiatives was included. Those that confirmed no radon testing were included in the ‘not tested for radon’ group. Regions that expressed interest in future testing but did not confirm current radon testing were included in the ‘interested in future testing’ sub-group in the ’not tested for radon’ group. All information was collated into brief summaries for each region. Historical testing initiatives were included.

## 3. Results

A cross-sectional representation of the number of schools testing for radon at the end of data collection in fall 2017 is illustrated in Figure 2.

Thirty-seven first level contacts were contacted across Canada. Their overall response rate was 65% (24/37). Response rate was lowest among contacts within the ministries of Health. The provinces with no first level response were Ontario, British Columbia, Manitoba, Alberta, Prince Edward Island and Newfoundland and Labrador. Newfoundland and Labrador was not included in Figure 1 to ensure anonymity.

Surveys were then sent to 246 second level contacts in British Columbia, Manitoba, Ontario, Alberta, Prince Edward Island and Newfoundland and Labrador where no first level responses were received by the end of the grace period. The response rate for second level contacts was approximately 26%. Third level contacts were composed of lung associations, teachers’ associations, ministries of Infrastructure and those who reached out to our research team independently to add clarity to our survey questions or to indicate alternative contacts that may lead to more accurate and up-to-date information.

Saskatchewan, New Brunswick, Nova Scotia, Prince Edward Island and Yukon reported that all schools have been tested for radon, although data were not disclosed to verify all of these reports. Only Prince Edward Island and Yukon make school radon testing results publicly available. In Quebec, in 2012, a pilot project testing 65 primary schools was conducted, and an aggregated report was made publicly available [20,21]. Following the release of this report, another radon testing program was launched that tested 57% of schools in Quebec for radon as a result of a collaborative approach led by the ministries of Education and Health (a product of the Inter-Sectoral Committee on Radon). British Columbia, Alberta, Manitoba, Ontario and Newfoundland and Labrador had low rates (below 20%) of radon testing in schools as of the date that this research was conducted (2017), and little information about this testing has been released.

Over 500 schools expressed interest in testing or had plans to test for radon in 2018, and the federal government continues to encourage radon testing in all schools in Canada. Alberta, Prince Edward Island and Nova Scotia indicated that policy and procedures have been developed to ensure that new schools include radon remediation rough-in systems, or include radon as a health risk in the construction and design discussions for new buildings to ensure that radon remediation systems can be easily installed if required.

Due to permafrost, many school buildings in Nunavut and some in the Northwest Territories are constructed above ground on piles. Further, these school buildings have space and airflow between the underside of the building and the grade below, allowing for radon to escape into the atmosphere instead of accumulating in the building above. Without direct contact with the ground, radon accumulation is unlikely; therefore, testing may not be required. Further investigation is required to explore radon levels in these types of buildings before policy should be considered.

In some cases, some contacts referenced the 2011 Radon Potential Maps of Canada created by industry professionals to justify why school districts did not need to test for radon [22]. Where low levels of radon were predicted on these maps, it was assumed testing was not needed. These maps were developed based on geological information of the soil composition across Canada, and they seek to predict potential radon emittance. However, the maps cannot predict indoor radon levels in buildings, such as in schools or homes. This raises the concern of the misuse of the maps’ information. It also highlights the need for consistent mapping methods to be developed by government that accurately explain indoor radon testing results of buildings, including schools. Transparency and accessibility to radon testing results of schools can demonstrate the variations that can exist in indoor environments within the same geographic area and be leveraged as a tool for decision makers.

Overall, our results showed that trends in radon testing in schools across Canada vary drastically between provinces and territories. Communication about radon testing between agencies, staff and parents was identified as a key challenge to implementing testing programs. It was also noted that the lack of national, and provincial or territorial government policy and resources, such as maps of radon measurements in buildings, contributed to inaction and further influenced individuals’ priority and perceived health risks of radon.

## 4. Discussion

Without a national school testing mandate, provinces approach the issue of radon testing in different ways. In Saskatchewan, New Brunswick, Nova Scotia and Yukon, the agencies responsible for testing included one of the following ministries: Health, Education, Infrastructure, Transportation and Labour. In British Columbia, Manitoba and Ontario, where there is no provincial or territorial government involvement, some regional school districts assumed the responsibility of testing and remediation on their own. These programs are often assisted and supported by external organizations, such as school board associations, regional health authorities and teachers’ unions.

Building codes across Canada now recommend that new government buildings, including schools, be designed and built with a radon mitigation ’rough-in’ system that can be activated if radon levels are above the Government of Canada’s radon guideline. However, the national building code is a model, and not enforceable in all provinces and territories [23,24], as the building code needs to be adopted by provinces and territories for enforcement purposes. Even with a radon mitigation ‘rough-in’ system, radon testing is required to determine if levels exceed the guideline and if completion of the ‘rough-in’ system is needed. Radon testing in schools continues to increase, with over 500 schools across Canada expressing interest or developing plans to test for radon in 2018 and beyond. Some of this action was a product of the research group’s outreach to gather data for this study, raising awareness about radon exposure in school environments among key stakeholders.

Despite best efforts, in Canada, knowledge of radon, the risks associated with exposure and the health effects remains low, with about 50% of Canadian households that had heard of radon, and only about 4% of households that had tested for radon [9]. Since the conclusion of this research in 2017, results have been shared with participants who contributed to data collection and other interested parties—knowledge users, experts, industry and other government networks—in an effort to raise awareness about radon exposure in schools. Additionally, we have used the results of this research to engage in meaningful dialogue with key stakeholders in order to support the development of consistent radon testing policies for schools across Canada.

Some knowledge gatekeepers and those who provided information contacted the research group following the completion of the study to reveal new testing in some provinces and territories. Since the advent of our study, an additional 106 schools were tested in British Columbia, including six school districts (92 schools) where testing was supported by members of regional health authorities. An additional 50 schools in Alberta were tested in fall 2018, and 10 additional schools in Manitoba were tested. School testing programs are underway in Newfoundland and Labrador and additional schools in British Columbia’s interior region. This study is a baseline summary of the number of schools testing for radon across Canada as of fall 2017, and it actively seeks new school radon testing information to augment these results.

Testing and any remediation to school buildings must be conducted by certified professionals. An active sub-slab depressurization system is considered the most effective action to take for radon mediation in homes, which involves installation of a pipe through the foundation and exiting out through the roof or side of the building. In larger buildings, such as schools, improvements to heating, cooling and ventilation systems can reduce radon levels. Heating, cooling and ventilation systems can influence indoor radon gas levels by pressurizing the building to drive radon out, or with ventilation that mixes with outdoor air to dilute indoor radon levels [25]. However, this is highly dependent on the design and regular maintenance of the building’s systems. Any successful radon reduction program requires clear remediation methods to reduce indoor radon levels, including the regular maintenance of building systems and regular monitoring to ensure systems are functioning properly.

This research also suggested a lack of accountability for the duty of radon testing in schools. Many participants pointed to other organizations that would need to be involved to create policy or projects that target testing in public schools. This highlights the lack of institutional ownership around coordinating and implementing testing. Clear jurisdictional assignment of radon within a policy portfolio and collaborative efforts would ensure testing is conducted regularly and effectively in all Canadian schools. This could include inter-ministerial coordination of efforts between ministries of Education, Health, Infrastructure (for example) as well as technical and building maintenance partners.

A national policy would ensure radon testing, mitigation and ongoing monitoring in all Canadian public schools. An inter-ministerial committee comprising the ministry of Education and Health in each province or territory would be responsible for policy implementation and may implore the support of other ministries and agencies as needed. Policy should include routine radon testing of all school buildings and also enforce remediation if radon test results are above the Government of Canada guideline (200 Bq/m^3^). The policy should emphasize radon re-testing following remediation to ensure effectiveness and suggest that radon testing be conducted biennially to ensure radon levels remain low in all schools. It is recommended that regular communication on radon testing and mitigation programs in schools include key stakeholders, parents, teachers and other school administrative staff. Radon should also be included as a health hazard topic for new school buildings and include requirements to test for radon post-construction. A comprehensive policy would also include communication strategies and radon awareness training to increase Canadians’ knowledge of radon exposure more broadly.

## 5. Conclusions

The results of this research can provide insights to inform policy around testing of radon in publicly funded schools across Canada. Participants have highlighted the need for transparent communication of radon risk, test results and mitigation costs between the government, school boards and parents. Testing of radon is the first step in identifying potential exposure levels. Remediation and follow up testing are fundamental to ensure that remediation efforts have reduced radon levels in buildings and that the overall risk of exposure has been lowered. A recommendation and measurement guideline from the Government of Canada specifies radon testing and mitigation within Canadian schools [10]. Our research group recommends that each region of Canada adopt these recommendation requirements as a mandated policy that could be incorporated into routine maintenance protocols for school buildings and include further training of appropriate personnel as certified radon testing and mitigation professionals. A collaborative, mandated policy can ensure that the radon exposure for children in Canadian schools is as low as possible.

## Figures and Tables

**Figure 1 ijerph-18-05469-f001:**
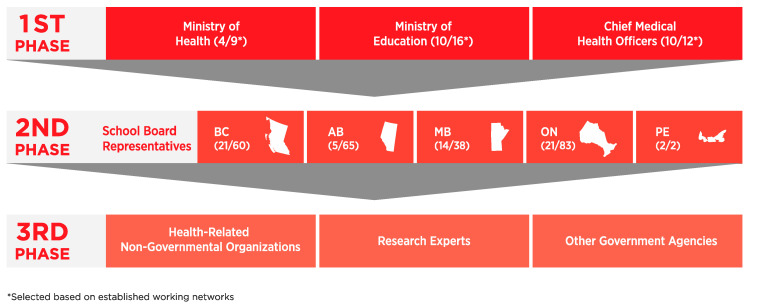
Data collection method and response rates for each level of knowledge gatekeepers.

**Figure 2 ijerph-18-05469-f002:**
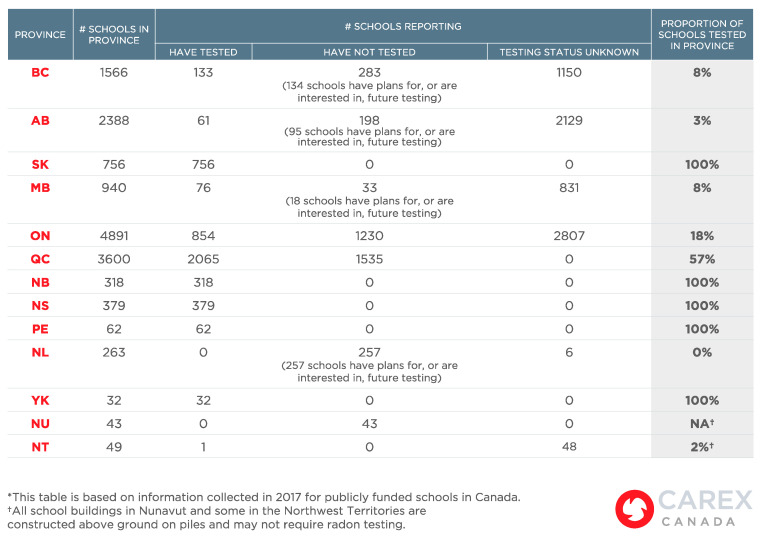
Summary of results by province for Canadian schools reporting radon testing.

**Table 1 ijerph-18-05469-t001:** Survey questions sent to gatekeepers.

	Question
1	How many schools were tested for radon and when? Additionally, please include any related details, what rooms were tested and available testing results.
2	Has there been any remediation where levels were reported above the current Canadian radon guideline? (If testing has been previously conducted).
3	If applicable, are there any current programs that monitor how often schools should follow up and test again for radon?
4	What was the level of engagement during testing and result communication by staff, teachers and parents?
5	How much was financially invested into the testing efforts by the province or school district?

## Data Availability

All policies referenced in this paper are already publicly accessible.

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
