# Peer review of "Radon in Schools: A Review of Radon Testing Efforts in Canadian Schools"

_ijerph, 2021, doi:10.3390/ijerph18105469_

Round 1

Reviewer 1 Report

This manuscript presents an interesting review of radon testing efforts in Canadian schools.

Overall it is well written and the results are clearly presented. It is interesting that both students and staff are mentioned in the text.

There are some comments detailed below:

- Maybe “preventing” in the title should be reconsider.

- Line 8: Number “3” should be in the same font-type that "1" and "2".

- At some point, preferably at the beginning of the manuscript, it should be mentioned that "radon" in the text refers to the radionuclide 222Rn. 

- Lines 57-58: Could you detail in the manuscript how is defined the threshold of 200 Bq/m3? Is it an annual average?

- Figure 2.- Do you think it could be interesting to show a total/summary file?

- Line 151: 64/248 is different from the data in Figure 1 (63/248). Is it due to the anonymity mentioned on line 147?

About Results: Interesting paradoxical observation regarding about the misuse of maps and the difficulty of explaining the true interpretation of their meaning.

Author Response

Wording and numerical suggestions have been incorporated. Additions regarding how Canada developed the 200 bq/m3 guideline were added. The totals are different and include the term "approximately" to reduce confusion around the anonymous data issues.

We are unable to include a summary file at this time. 

Reviewer 2 Report

The manuscript entitled ‘‘Preventing radon in schools: A review of radon testing efforts in Canadian schools‘‘ is a good scientific paper. I usually list few major corrections to be reviewed along with minor corrections when I review papers. But I only noted the next six minor improvements that could be addressed prior to modification. This paper should be published as it can provide insights to inform policy around testing of radon in publicly-funded schools across Canada. It is a good paper.

Here are some minor problems that should be improved.

Lines 85-86: It is mentioned: Finland, Norway and Switzerland are examples of countries that enforce 85 legal limits for radon levels within school buildings. You could cite these thresholds in schools as a comparative tool.

Line 92-93: Less attention has been paid to school environments. The. Do you have a hypothesis why there is less attention paid to school environments in Canada?

Table 1 – question 4: I suggest that the gatekeepers could rate the level of engagement during testing with a result between 0 and 10, 0 being no engagement and 10 being a complete engagement. It will ease the compilation of the results and the answering from gatekeepers. This is just a comment as the survey was already sent to the gatekeepers…

Line 158-159: In Quebec, in 2012, a pilot project testing 65 primary schools was conduct. was conducted.

Line 160: another radon testing program was launch. was launched

Line 168-169: Radon remediation systems included in the policies and procedures of Alberta/PEI/NS are active or passive depressurization systems? …it is explained at lines 207-209.

Author Response

Reference levels for EU countries added to the paper. We expanded the section on the jurisdictional challenges of addressing schools and added more context to the confusion that exists about where the responsibility lies regarding radon testing and remediation within the government itself. All grammatical and textual edits were incorporated.  

Reviewer 3 Report

Line 79-80: The European Council Directive 2013/59 / Euratom of 5 December 2013 established basic safety standards for protection against the dangers arising from exposure to ionizing radiation.

Line 159: was conducted

Line 160: was launched

Line 167: radon testing

Author Response

We clarified the EU directive document and have added this as a reference. All other editorial changes were made. Thank you. 

Reviewer 4 Report

I read the article with great interest.
The research problem is very interesting, the presentation of the research
and results is excellent. The work is based on 24 references, among which
there are several recent ones. I believe that the work raises a very
important issue of children's exposure to radon, one of the modifiable
factors in the development of lung cancer. Preventing this exposure,
also by spreading the knowledge about the problem and drawing attention
to national governments through such research, is of importance that
cannot be overestimated when it comes to protecting children from the
effects of radon.

Author Response

Thank you for your comments!

Reviewer 5 Report

The radon measurements in school is a very important question nowadays. The manuscript is a well-written, detailed description of the data collection and "the sampling" before the measurement. 

Despite all that I can not accept this manuscript because it is not reach the quality of the journal. 

My recommendation is the improving the manuscript with the data from the radon survey and in that case, it will be acceptable. 

Author Response

Currently, there is no survey of radon measurements taken in schools. If this data did become available, it would be a tremendous addition to the current radon measurement initiatives. It would be interesting to see who would juridictionally take on a national survey of radon in schools. 

Reviewer 6 Report

This was an excellent review paper regarding radon in schools of Canada. It was a pleasure to read it. Since working with radon for many years, the paper was of grate interest. More specifically, it emphasises the very sensitive subject of radon in schools. No particular item was addressed despite reading it very carefully. 

I definitely suggest its publication in the present form.

Author Response

Thank you for your comments!

Round 2

Reviewer 5 Report

I have no comments